# Patients with Adolescent Idiopathic Scoliosis Have Higher Metabolic Cost during High-Intensity Interval Training

**DOI:** 10.3390/ijerph20032155

**Published:** 2023-01-25

**Authors:** Rufina Wing-Lum Lau, Rachel Lai-Chu Kwan, Jack Chun-Yiu Cheng, Stanley Sai-Chuen Hui, Tsz-Ping Lam

**Affiliations:** 1School of Medical and Health Sciences, Tung Wah College, Hong Kong SAR, China; 2SH Ho Scoliosis Research Lab, Joint Scoliosis Research Center of the Chinese University of Hong Kong and Nanjing University, Department of Orthopaedics & Traumatology, Faculty of Medicine, The Chinese University of Hong Kong, Hong Kong SAR, China; 3Department of Sports Science and Physical Education, Faculty of Education, The Chinese University of Hong Kong, Hong Kong SAR, China

**Keywords:** adolescent idiopathic scoliosis, exercise, high-intensity interval training, metabolic cost, oxygen consumption

## Abstract

Background: Patients with adolescent idiopathic scoliosis (AIS) are found to have a lower level of physical activity, and may have reduced exercise capacity due to spinal deformity. Previous study showed the benefits of high-intensity interval training (HIIT), named E-Fit, which is specifically designed for patients with AIS to improve musculoskeletal health and psychological well-being. To optimize the beneficial effects of training, the current study aimed to investigate the appropriate exercise intensity and metabolic demand in patients with AIS when performing E-Fit. Methods: In all, 22 female subjects, 10 diagnosed with AIS and 12 gender-matched healthy controls, aged between 10 and 16 years, were recruited. Subjects were instructed to perform two trials of a seven min E-Fit. Breath-by-breath gas exchange parameters including oxygen consumption (VO_2_), heart rate (HR) and the rate of perceived exertion (PRE) were measured during exercise. Demographic data and clinical features of AIS and body composition were obtained. Metabolic demand between AIS and control groups was compared using MANOVA with covariates adjustment. Results: Patients with AIS had an earlier onset of menarche (*p* = 0.01), higher visceral adipose tissue (*p* = 0.04) and percentage body fat (*p* = 0.03) as compared to controls. Patients with AIS showed a significantly higher adjusted means of VO_2_ average in both the first (*p* = 0.014) and second trials (*p* = 0.011) of E-Fit. The adjusted mean of the highest measured VO_2_ was higher than healthy controls and reached statistical significance in the second trial (*p* = 0.004). Both the AIS and control group exercised at a similar percentage of VO_2_ peak (64.26% vs. 64.60%). Conclusion: Patients with AIS showed higher oxygen consumption during E-Fit than heathy controls, which might indicate a higher metabolic cost. Patients with AIS could carry out exercise at a moderate exercise intensity similar to that of healthy controls, but special considerations in designing an exercise program, such as frequent rest intervals, would be useful to avoid fatigue among patients with AIS.

## 1. Introduction

Adolescent idiopathic scoliosis (AIS) is a three-dimensional deformity of the spine, mostly occurring in females between the age of 10 and 16 years, with a prevalence of 1–4% worldwide [1,2]. Complex spinal deformity in AIS is thought to be associated with the differential development of the musculoskeletal system [3], which could influence the cardiorespiratory system, leading to a lower physical capacity even in patients with mild curve [4,5,6]. Reduced physical capacity in patients with mild-to-moderate AIS could result in an increase in energy expenditure for performing physical activities, and might have a detrimental effect on pubertal development carrying over into adulthood [7,8].

The developmental deformation of the spine and ribcage in AIS could influence cardiorespiratory functions. Recent study revealed that patients with moderate AIS had an imbalance growth between the sternum and thoracic spinal column [9], with shallow and flat chest ribcage rotation [10,11]. The decreased mobility in the spine and ribcage of AIS affects the ability of the chest wall, abdomen, and diaphragm, to expand when breathing [12]. These restrictions could cause an increase in breathing frequency and a decrease in oxygen consumption [13]. Apart from structural abnormalities, chronic physical deconditioning from the lack of regular exercise, reduced ventilatory capacity, and generalized muscle dysfunction, could lead to a loss of exercise capacity [4,14,15,16,17]. Patients with moderate AIS exhibited respiratory muscle weakness and impaired breathing pattern, even during walking [17]. Nutritional status, physical activity level, and systemic inflammation were thought to attribute to generalized deranged muscle functions in patients with AIS [16]. Patients with AIS who engaged in regular aerobic exercises were shown to have significant improvement in pulmonary functions [6] and aerobic capacity [18]. In patients with severe AIS, who had curves that progressed up to surgical thresholds [19], recent study showed that combining resistance and aerobic training improved functional exercise capacity and pulmonary functions, more than a training regimen with aerobic training alone.

An E-Fit exercise intervention (E-Fit), which is a high-intensity interval training (HIIT) specifically designed for patients with AIS, was developed in order to promote habitual exercise. E-Fit comprises of whole-body aerobic and resistance exercises, with frequent rest intervals. Previous study showed that E-Fit could provide physical benefits on bone mineral density and muscle endurance, as well as the psychosocial benefits of improving sport participation, and self-image in patients with AIS [20]. Metabolic stress induced by exercise intervention can vary between individuals and therefore it is important to determine the appropriate exercise intensity, which could result in better adherence and effectiveness [21]. This prospective case-control study aimed to determine the metabolic demand of E-Fit by evaluating the cardiovascular responses in patients with AIS, compared with healthy controls.

## 2. Materials and Methods

### 2.1. Subjects

Twenty-two female subjects between the ages of 10 and 16 were recruited in this study. 10 subjects who had a diagnosis of AIS confirmed by clinical examination and standard standing posteroanterior radiograph of the whole spine, Cobb angle of ≥15°, without prior treatment were recruited from a local hospital. Exclusion criteria included: Cobb angle of ≥40°; scoliosis with any known aetiology, endocrine and connective tissue abnormalities; heart condition or other diseases that could affect the safety of exercise; eating disorders or gastrointestinal malabsorption disorders; and currently taking medication that affects bone or muscle metabolism. Twelve healthy female adolescent subjects matched in age without spinal deformity were recruited as controls. Written informed consent was obtained from all subjects and their guardians before enrolment. The study was approved by the local ethics committees in compliance with the Helsinki Declaration.

### 2.2. Baseline Measurements

Anthropometric measurements were measured with standard stadiometry techniques. Sexual maturity was assessed by self-reported onset of menarche and standard Tanner Scale using the validated protocols [22]. For AIS group, skeletal age was measured with Thumb Ossification Composite Index (TOCI) (stage 1–8 with higher values representing more mature bone) using hand X-ray [23]. Body composition parameters, including fat-free mass (FFM), body fat percentage (BF%) and visceral adipose tissue (VAT), were examined using the bioelectrical impedance analysis (BIA) (InBody 720; Biospace Co., Ltd.; Seoul, Republic of Korea). The average physical activity level in the past year was assessed using a self-administered and validated 1-item questionnaire, The Chinese University of Hong Kong: Physical Activity Rating for Children and Youth (CUHK-PARCY) [24].

### 2.3. Testing Procedures

Subjects in both groups were instructed about the E-Fit exercises. The room for obtaining outcome measures had an ambient temperature between 22–24 °C with approximately 50% humidity. Each subject was required to undergo a submaximal treadmill exercise testing and performed two trials of E-Fit. Each trial of E-Fit lasted for around 7 min in total and included 12 short and intense exercises. Selected E-Fit exercises in each trial had included a mix of aerobic and resistance exercises involving large muscle groups of the whole body, such as elbow-to-knee, arm walk, triceps dips, bridging, squat with weight and squat thrust. Subjects were given 10 min rest period between each testing. A 50% of heart rate reserve (HRR) was first determined by maximal heart rate (HRmax) defined as 200 beats per minute (bpm), and resting heart rate (HRrest) measured after a rest period of 10 min.

Prior to the submaximal treadmill exercise testing, subjects were first instructed to have a warm-up and then fast walk at a self-selected pace to obtain a heart rate (HR) of 50% HRR within a 3 min time. The main part of the exercise testing involved normal walking at comfortable pace for 3 min, fast walking for another 3 min, then fast walking at the self-selected pace for 6 min until a steady HR was reached, with the HR of 5th and 6th min not exceeding 6 bpm. Subjects proceeded to cool down and walk at a comfortable pace for 5 min until HR returning below 90 bpm. Subjects were seated immediately after completing the exercises and measurements were continued for a further 2 min.

### 2.4. Outcome Measures of Oxygen Consumption (VO_2_), Heart Rate (HR) and Rate of Perceived Exertion (RPE)

A telemetric portable gas analyzer system (Cosmed K5; Cosmed Srl, Rome, Italy) was utilized to obtain breath-by-breath measurement of cardiorespiratory and gas exchange parameters to analyze oxygen consumption (VO_2_) expressed as mL/kg/min during the E-Fit [25]. Heart rate (HR) was measured continuously in conjunction with the gas analyzer system using a chest strap (Garmin HRM3-SS; Garmin, Olathe, KS, USA). Breath-by-breath values were averaged over 30 s interval of each movement element. The system was calibrated for volumes and gas exchange composition using reference gases of known concentrations. Respiratory gas measurements were taken during the submaximal exercise testing using a motorised treadmill (GK2200; Mobility Research, Tempe, AZ, USA) and two trials of E-Fit. Modified Borg Scale was used to assess the subjective rating of perceived exertion at the beginning and immediately after the end of each E-Fit trial. It has shown to be a reliable indicator for exercise exertion with a good accuracy in children older than 9 years of age [26,27].

### 2.5. Statistical Analysis

Independent *t*-test and chi-square test were used to compare the demographic data and baseline outcome variables between AIS and healthy control groups. Multivariate analysis of variance (MANOVA) was performed to compare the metabolic demand of E-Fit between groups. Potential confounders such as onset of menarche, VAT, BF% and resting VO_2_ were included in the model where appropriate. SPSS statistic software (version 26; SPSS Inc., Chicago, IL, USA) was used for statistical analyses. *p*-value of < 0.05 was adopted as the general level of significance. Forty female adolescents aged between 11 and 14 years with AIS were targeted and recruited from an out-patient scoliosis clinic in a local hospital. All subjects would be included if they were female AIS patients who were newly diagnosed with AIS by standard standing long X-ray examinations, with Cobb angle larger or equal to 15°, without prior treatment for their AIS and have been cleared for physical activity by their doctors. Subjects were excluded if they had: (i) Cobb angle larger or equal to 40°; (ii) scoliosis with any known aetiologies, such as congenital, neuromuscular, metabolic and skeletal dysplasia; (iii) known endocrine and connective tissue abnormalities; (iv) known heart condition or other diseases that could affect the safety of exercise; (v) eating disorders or gastrointestinal malabsorption disorders; and (vi) currently taking medications that affecting their bone or muscle metabolism. All subjects signed a consent form in the presence of their parents after thorough explanation by the attending doctor.

## 3. Results

### 3.1. Group Comparison

Ten patients with AIS and 12 age-matched healthy controls with a mean age of 12.83 ± 2.09 years old completed the study. Both groups had similar anthropometric measurements, but the AIS group had an earlier onset of menarche (*p* = 0.014), higher visceral adipose tissue (*p* = 0.037) and BF% (*p* = 0.027). Both groups did not show any statistical significance in physical activity level (Table 1).

### 3.2. Metabolic Demand between AIS and Healthy Controls

The resting HR, 50% HRR and predicted VO_2_ peak in both groups did not show significant difference, however, the AIS group had a statistically significant lower resting VO_2_ at 3.19 mL/kg/min (*p* = 0.045) than the control group at 4.16 mL/kg/min (Table 2).

During E-Fit, the AIS group also showed significant higher adjusted means of VO_2_ average in the first trial (20.27 mL/kg/min, *p* = 0.033). The adjusted mean of the highest measured VO_2_ was numerically higher than healthy controls, reaching statistical significance in the second trial (37.46 mL/kg/min vs. 33.87 mL/kg/min, *p* = 0.010) with a marginally significant overall-between effect (*p* = 0.055). However, the AIS group was observed to have a significantly lower VO_2_ average than the healthy controls in the second trial (19.45 mL/kg/min vs. 19.48 mL/kg/min, *p* = 0.022) with a reducing trend in HR peak, HR average and RPE. The overall within-group effect of RPE was statistically different (*p* = 0.016) with the AIS group showing a significant decline in RPE score from 2.85 to 2.55 (adjusted mean of 2.70 to 2.43) towards the second trial of E-Fit. Both the AIS and control groups exercised at a similar percentage of VO_2_ peak (62.85% to 64.23% vs. 62.44% to 65.94%) in both trials. The BP and SpO_2_ at pre-test (measured after a 10 min rest period before each trial) and post-test also showed no significant difference between the groups. The metabolic demand parameters between the AIS and control group when performing E-Fit are presented in Table 3.

## 4. Discussion

This was the first study to investigate the metabolic demand of patients with AIS performing E-Fit, which is an exercise intervention utilizing HIIT. HIIT has been shown to be a feasible, time efficient and efficacious strategy for improving physical and psychological well-being in healthy adolescents [28] and patients with AIS [20]. The significantly lower resting VO_2_ in the AIS group found in this study indicates that patients had a lower resting metabolic demand in comparison to healthy controls. Multiple factors were shown to be related to a lower resting metabolic demand, including higher body fat, lower bone density, reduced skeletal muscle, nutritional status and the quality of sleep [29,30,31]; these factors were also associated with AIS [32]. The cause of lower resting VO_2_ found in the AIS group of this study is unclear, but they were found to have a higher VAT and BF%.

Both the AIS and control groups showed no difference in the predicted VO_2_ peak, which could imply both groups showed similar peak aerobic capacity. This finding was in accordance with those reported by Czaprowski et al., that the maximum oxygen consumption calculated based on a submaximal physical test did not differ between patients with mild AIS with a Cobb angle of less than 25°, and controls, during a cycle ergometer test [8]. Bas et al. [33] and Leech et al. [34] also found similar maximum oxygen consumption between patients with mild AIS and healthy controls. In addition, both the AIS and control groups exercised at a moderate-intensity level, with a similar percentage of VO_2_ peak (62.85% to 64.23% vs. 62.44% to 65.94%) in both trials. The comparable predicted VO_2_ peak in both groups indicated that there was no exercise limitation in patients with AIS, in comparison to healthy controls. E-Fit also seemed to have an appropriate and adequate training intensity for both groups, at a moderate level.

The AIS group showed a numerically higher adjusted mean of the highest measured VO_2_ and reached statistically between-group significance in the second trial, indicating that they demonstrated higher oxygen consumption and increased metabolic cost when performing E-Fit. Mahaudens et al. also concluded that patients with mild AIS had an increased metabolic cost by around 30%, and a decrease in muscle efficiency even in a common activity such as walking [35]. They suggested that the increased physical effort exerted in patients with AIS was not only due to a mechanical disorder but also inefficient muscle or muscular dysfunction. Kearon et al. reported that patients with AIS showed reduced strength in both respiratory and limb muscles, which was associated with a decrease in calculated lean mass [15]. Martínez-Llorens et al. also found patients with AIS had generalized muscle dysfunction including respiratory muscles, upper and lower limb muscles on both sides, which contributed to reduce exercise capacity [16]; however, the causes of muscle dysfunction were not fully understood. Several factors, such as nutritional status, physical activity level, reduced bone density, melatonin-level related to sleep quality or systemic inflammation, could influence the systemic mechanisms of muscle function [3,16,36].

It is worth noting that the AIS group showed a drop in HR peak, HR average and RPE in the second E-Fit trial (Figure 1). They had a significantly higher adjusted mean of the highest measured VO_2,_ but a significantly lower adjusted mean of VO_2_ compared with the healthy controls. This finding and observation suggested that patients with AIS might be showing signs of fatigue during the second E-Fit trial. Some patients with AIS were observed to have slowed down and it became more difficult for them to follow the exercise rhythm, despite similar verbal cueing given to both groups. Previous studies also reported high scores of perceived exertions in patients with AIS during exercise [6,37]. Patients with AIS were observed to have shallow breathing, increased breathing frequency and an altered breathing pattern of disproportion between inspiration and expiration [8,13]. Shneerson et al. found patients with AIS, on average, hyperventilated by 20% more than healthy controls [38]. Decreased cardiovascular fitness in patients with AIS could affect tidal volume, leading to an increase in the respiratory rate, compensating for a larger oxygen demand [39]. Apart from an observed reduction in spinal and chest wall mobility, this could be due to the weakness of respiratory muscles and chronic deconditioning [8,17]. We postulated that the increased physical effort and reduced exercise capacity observed in the current study were likely due to muscle dysfunction, such as reduced overall muscle strength, rather than cardiorespiratory restriction.

Systematic training, such as E-Fit, could induce metabolic and structural adaptations to improve the function and endurance of respiratory muscles [40,41]. Although exercise training may have a limited effect on the maximum pulmonary function, it can reduce the feeling of breathlessness and improve the ability to sustain high levels of submaximal ventilation during exercise [18]. Special considerations should be made when designing an exercise program for patients with AIS. For example, slower exercise rhythm with larger movements, more frequent rest intervals and adding more cueing could help prevent early fatigability, and sustain exercises at an adequate intensity and duration.

A limitation of this current study was that we only recruited a small sample of female subjects, of single ethnicity, with mild-to-moderate curve severity. Future studies could recruit more subjects with different ethnicity, a wider age range, various curve types and severity, for stratification analysis. The physical activity level between the AIS and control groups was assessed using a self-reported questionnaire, which is useful to differentiate the different levels of physical activity in our subjects, and to identify subjects with a high level of physical activity at baseline. However, self-reported questionnaires rely on subjects’ recall ability and can be less robust in assessing in detail the subject’s physical effort and energy expenditure. Further, the VO_2_ max was not measured directly in the current study; however, a submaximal treadmill exercise test was utilized to predict the peak of oxygen consumption. Submaximal tests, including the measurement of respiratory gas exchange, are widely accepted in the evaluation of physical capacity in children and adolescents, with or without AIS [7,15,42]. Since this study focused on the metabolic demand of patients with AIS performing E-Fit, assessments on respiratory functions and muscle strength were not included. The relationships between other possible underlying causes, such as respiratory functions or muscle functions, that might influence the reduced exercise capacity as observed in the current study, were beyond the scope of this study but are worthy of consideration when further studies are planned in this regard.

## 5. Conclusions

The current study indicated that patients with mild-to-moderate AIS had higher oxygen consumption and metabolic cost during E-Fit, as compared with healthy controls. Patients with AIS could carry out exercise at a moderate exercise intensity level, similar to that of healthy controls, but special considerations in designing the exercise program would be useful to avoid early fatigability. Further studies are warranted to investigate idiopathic scoliosis at various stages of the disease for defining the applicability of E-Fit for those with early onset, or those with curves reaching surgical threshold, and for evaluating the relationship between cardiorespiratory and muscle functions, to unravel the underlying causes of limited exercise capacity in patients with AIS.

## Figures and Tables

**Figure 1 ijerph-20-02155-f001:**
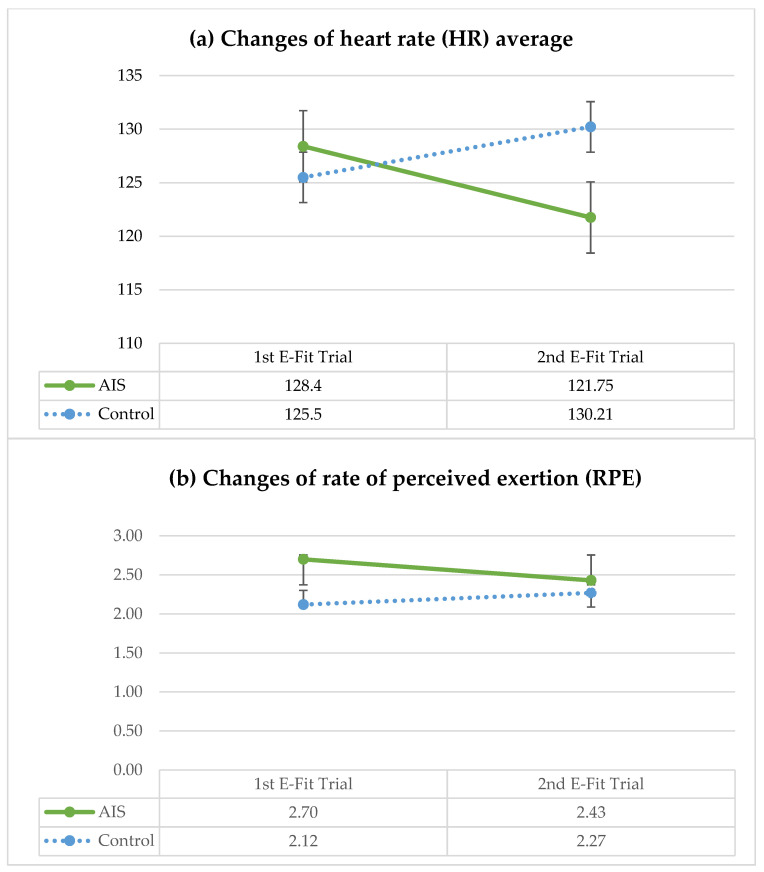
Changes of: (**a**) heart rate (HR) average; and (**b**) rate of perceived exertion (RPE), between AIS and control groups in both E-Fit trials.

**Table 1 ijerph-20-02155-t001:** Basic characteristics between AIS and control groups.

	AIS (*n* = 10)	Control (*n* = 12)	*p*-Value
Age (yr) ^a^	13.71 ± 2.04	12.11 ± 1.92	0.080
Anthropometric measurements			
Body weight (kg) ^b^	44.38 ± 6.67	42.00 ± 8.39	0.477
Body height (cm) ^b^	154.57 ± 7.14	152.67 ± 7.96	0.565
Sitting height (cm) ^a^	78.50 ± 16.07	86.33 ± 14.63	0.668
Arm span (cm) ^b^	153.90 ± 9.40	153.98 ± 7.33	0.446
BMI (kg/m^2^) ^a^	18.61 ± 1.27	17.86 ± 2.12	0.467
BMI by arm span ^a^	18.63 ± 1.40	18.31 ± 2.15	0.684
Maturity			
Tanner scale—breast development (1–5) ^c^	3.20 (3, 4)	2.25 (2, 3)	0.312
Tanner scale—pubic hair development (1–5) ^c^	3.10 (3, 4)	1.92 (1, 3)	0.155
Menarche onset (months) ^a^	22.90 ± 15.28	7.83 ± 16.28	0.014 *
Curve Features			
Cobb angle (˚)	24.50 ± 6.38	N/A	N/A
TOCI (stage 1–8)	5.60 ± 1.71	N/A	N/A
Body Composition ^b^			
Fat free mass (kg)	32.69 ± 3.97	33.32 ± 5.53	0.768
Skeletal muscle mass (kg)	17.18 ± 2.25	17.49 ± 3.39	0.806
Visceral adipose tissue (cm^2^)	52.67 ± 18.28	38.19 ± 11.89	0.037 *
BF%	24.78 ± 4.16	19.99 ± 5.10	0.027 *
Physical Activity Level			
CUHK-PARCY ^c^	4.00 (3, 6)	5.33 (6, 8)	0.698

The values are presented as mean ± standard deviation or median (range) as appropriate. ^a^: Student’s *t*-test was used on data with normal distribution. ^b^: Mann-Whitney U test was used on data which is not normally distributed. ^c^: Chi-square test was used for the dichotomous data. Abbreviations: BMI = body mass index; TOCI = thumb ossification composite index; BF% = body fat percentage; CUHK-PARCY = The Chinese University of Hong Kong: Physical Activity Rating for Children and Youth. * *p* ≤ 0.05.

**Table 2 ijerph-20-02155-t002:** Baseline metabolic demand parameters between AIS and control groups.

	AIS (*n* = 10)	Control (*n* = 12)	*p*-Value
Resting HR (bpm)	89.10 ± 11.41	89.75 ± 9.54	0.190
50% HRR (bpm)	144.55 ± 5.70	143.67 ± 3.87	0.367
Resting VO_2_ (mL/kg/min)	3.19 ± 0.54	4.16 ± 1.03	0.045 *
Predicted VO_2_ peak (mL/kg/min)	57.43 ± 14.14	56.65 ± 11.57	0.353

The values are presented as mean ± standard deviation. Covariates: onset of menarche, body fat percentage and visceral adipose tissue. Abbreviations: HR = heart rate; 50% HRR = 50% heart rate reserve; VO_2_ = oxygen consumption. * *p* ≤ 0.05.

**Table 3 ijerph-20-02155-t003:** Mean, adjusted mean, standard deviation and standard errors for metabolic demand between AIS and control groups when performing E-Fit trials.

		AIS (*n* = 10)		Control (*n* = 12)		*p*-Value ^a^
		M ± SD	M_adj_ ± SE	M ± SD	M_adj_ ± SE	
HR peak	1st E-Fit Trial	152.30 ± 10.64	149.85 ± 3.96	145.58 ± 12.44	147.63 ± 3.55	0.204
	2nd E-Fit Trial	148.90 ± 17.01	143.78 ± 5.25	146.17 ± 14.01	150.44 ± 4.71	0.324
HR average	1st E-Fit Trial	131.20 ± 9.31	128.4 ± 3.13	125.50 ± 10.21	127.83 ± 2.81	0.093
	2nd E-Fit Trial	126.60 ± 11.65	121.75 ± 3.66	126.17 ± 11.67	130.21 ± 3.27	0.129
Highest measured VO_2_	1st E-Fit Trial	34.65 ± 7.18	36.85 ± 2.14	34.38 ± 6.12	32.54 ± 1.92	0.186
	2nd E-Fit Trial	35.24 ± 5.35	37.46 ± 1.20	35.72 ± 3.94	33.87 ± 1.08	0.010 *
VO_2_ average	1st E-Fit Trial	19.03 ± 3.14	20.27 ± 0.79	19.66 ± 2.46	18.63 ± 0.71	0.033 *
	2nd E-Fit Trial	18.73 ± 2.57	19.45 ± 0.56	20.08 ± 1.17	19.48 ± 0.51	0.022 *
%VO_2_ peak	1st E-Fit Trial	62.89 ± 16.97	62.85 ± 5.50	62.41 ± 13.98	62.44 ± 4.93	0.548
	2nd E-Fit Trial	64.98 ± 19.05	64.23 ± 6.06	65.32 ± 14.08	65.94 ± 5.43	0.692
RPE	1st E-Fit Trial	2.85 ± 0.35	2.70 ± 0.42	2.00 ± 0.36	2.12 ± 0.38	0.236
	2nd E-Fit Trial	2.55 ± 0.40	2.43 ± 0.36	2.17 ± 0.29	2.27 ± 0.32	0.143
	Overall within-group effect of RPE: *p* = 0.016 *					

The values are presented as mean ± standard deviation or adjusted mean (standard error) as appropriate. ^a^: *p*-value of adjusted mean. Covariates: onset of menarche, body fat percentage, visceral adipose tissue, and resting oxygen consumption. Abbreviations: HR = heart rate; VO_2_ = oxygen consumption; RPE = rate of perceived exertion. * *p* ≤ 0.05.

## Data Availability

The data presented in this study are available on request from the corresponding author. The data are not publicly available due to privacy issues of the subjects.

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
