# Peer review of "Patients with Adolescent Idiopathic Scoliosis Have Higher Metabolic Cost during High-Intensity Interval Training"

_ijerph, 2023, doi:10.3390/ijerph20032155_

Round 1

Author Response

We are grateful for your comments and suggestions. Please refer to the attached document for our responses and revision. Thank you for your valuable time in reviewing our manuscript. 

Reviewer 2 Report

First of all, congratulations for a well written paper.

There only few points that need to be better defined, in my opinion.

In the introduction section, all the pulmonary and functional implications of scoliosis that authors cite are correct, but they are more frequent and more clinically significant in severe scoliosis patients and much less frequent and less clinically significant in mild scoliosis cases. This should be better defined in the introduction section, which otherwise seems a little bit misleading.

Again, it would be crucial to have a stratification of the results based upon the severity of the main curve, since patients with a 15° curve may perform very differently from patients with a 35° curve. Moreover, a distinction based on the location of the curve (thoracic vs lumbar) seems to be appropriate.

The main limitation of the study, in my opinion, is the fact that CUHK-PARCY score was used to assess the baseline physical activity level of the cohort. The score is just based on the difference between light, moderate and vigorous activity, but it is not able to distinguish the different kind of aerobic effort that these activities may have. For example, “vigorous activities” include vigorous basketball competitions, vigorous soccer competitions and skiing, but these 3 activities, despite being all vigorous, are based on very different type of physical effort (basketball is more like jumping and fast running for small distances, while football is more like running for longer distances, skiing doesn’t require running at all, but is more based on lower limbs strength). Again, “moderate activities” include cycling and basketball shooting or generical “playground playing”, with obvious differences between the three. In that sense, patients with the same score, may have a very different performance in high intensity exercises, leading to a potential bias in the results of this study, which should be acknowledged in the limitations section.

Once these changes are made, this well written and interesting article is worthy of publication, in my opinion.

Author Response

We are grateful for your in-depth review and comments. Please find the attached file with a summary of our responses and changes made in the manuscript. Thank you for your time in reviewing our manuscript. 

Reviewer 3 Report

Thanks to the authors for their effort for putting together the present manuscript. The topic is interesting and its application can become useful. The manuscript has a very clear architecture and is well written.

I suggest acceptance of the present work under minor revision, as following:

- I'm not feeling confident about the CUHK-PARCY questionnaire: only 1 item to evaluate physical activity seems to me a little vague. I'd add a statement in the discussion section about the limitation that this gives.

- I don't think that reference 2 is correct, it cites a paper regarding early scoliosis and not AIS. I'd change that.

- As a scientific reader I would have liked to read more about the kind of exercise present in the Efit. I've seen the authors cited a previous work where all this is explained, but I think that adding some more information could be more pleasant to read.

Thanks for your work

Author Response

Thank you for your time in reviewing our manuscript. We really appreciate your insightful comments. Please find the attached for a summary of our responses and revision made in the manuscript. 
